# Investigating the Key Trends in Applying Artificial Intelligence to Health Technologies: A Scoping Review

TAWIL Samah[1,2]*, MERHI Samar[3]

1 Gilbert and Rose-Marie Chagoury School of Medicine, Lebanese American University, Beirut, Lebanon, 2 Institut National de Santé Publique d'Épidémiologie Clinique et de Toxicologie-Liban (INSPECT-LB), Beirut, Lebanon, 3 Faculty of Nursing and Health Sciences, Notre Dame University-Louaize (NDU), Zouk Mosbeh, Lebanon

* samah.tawil@lau.edu.lb (ST)

## Abstract

### Background

The use of Artificial Intelligence (AI) is exponentially rising in the healthcare sector. This change influences various domains of early identification, diagnosis, and treatment of diseases.

### Purpose

This study examines the integration of AI in healthcare, focusing on its transformative potential in diagnostics and treatment, and the challenges and methodologies. shaping its future development.

### Methods

The review included 68 academic studies retracted from different databases (WOS, Scopus and Pubmed) from January 2020 and April 2024. After careful review and data analysis, AI methodologies, benefits and challenges, were summarized.

### Results

The number of studies showed a steady rise from 2020 to 2023. Most of them were the results of a collaborative work with international universities (92.1%). The majority (66.7%) were published in top-tier (Q1) journals and 40% were cited 2–10 times. The results have shown that AI tools such as deep learning methods and machine learning continue to significantly improve accuracy and timely execution of medical processes. Benefits were discussed from both the organizational and the patient perspective in the categories of diagnosis, treatment, consultation and health monitoring of diseases. However, some challenges may exist, despite these benefits, and are related to data integration, errors related to data processing and decision making, and patient safety.

**Data availability statement:** All relevant data are within the paper and its Supporting Information files.

**Funding:** The author(s) received no specific funding for this work.

**Competing interests:** The authors have declared that no competing interests exist.

## Conclusion

The article examines the present status of AI in medical applications and explores its potential future applications. The findings of this review are useful for healthcare professionals to acquire deeper knowledge on the use of medical AI from design to implementation stage. However, a thorough assessment is essential to gather more insights into whether AI benefits outweigh its risks. Additionally, ethical and privacy issues need careful consideration.

## 1. Introduction

Artificial Intelligence (AI) englobes computational technologies that replicate processes associated with human intelligence, including thought, deep learning, adaptation, engagement, and sensory understanding [1,2]. Certain devices utilizing an interdisciplinary approach, such as robotic surgical systems [3], holographic and hybrid high-resolution Magnetic Resonance [4], or VBrain which is an AI-assisted brain tumor auto-contouring tool [5] are employed across various fields, particularly in medicine and healthcare [6]. These devices are capable of performing tasks that traditionally require human interpretation and decision-making [7,8]. The integration of AI in medicine dates back to the 1950s, with early attempts by physicians to enhance diagnoses through computer-aided programs [9]. Recent years have witnessed a surge in interest and advancements in medical AI applications, driven by the significantly enhanced computing power of modern computers and the abundance of digital data [10,11]. AI is rapidly transforming the landscape of medicine and healthcare, offering innovative solutions to various challenges [12,13]. It is progressively reshaping medical practices, offering diverse applications in clinical, diagnostic, rehabilitative, surgical, and predictive realms worldwide [14–16]. Researchers have utilized AI technology across a range of medical conditions, including detecting diabetic retinopathy [17], analyzing heart abnormalities [18], and predicting risk factors for cardiovascular diseases [19]. Furthermore, deep learning algorithms have been applied to pneumonia detection using chest radiography, achieving a sensitivity of 96% and a specificity of 64%, compared to radiologists, who demonstrated sensitivity and specificity rates of 50% and 73%, respectively [20]. These studies collectively demonstrate the diverse applications and transformative impact of AI in advancing medical research and patient care. With ongoing developments and increasing adoption, universities in the Middle East and North Africa (MENA) region acknowledged the vital importance of adapting to new technologies. They recognized the key role in transforming the region and advancing to the forefront of the digital economy and healthcare. Similarly, Lebanon is actively embracing AI tools marking significant strides in its technological landscape [21–23]. Recently, Lebanon announced the adoption of its first-ever AI Policy, underscoring a commitment to harnessing AI's potential for societal benefit [24–26]. Moreover, there is a growing interest in AI education, with a guide for becoming AI certified in Lebanon, providing insights for enthusiasts and beginners [27,28]. Although the Lebanese advancement in AI technology

was slowed by many factors such as COVID-19 pandemic and economic crisis [29–31], many Lebanese universities have introduced digital management systems and created new instructional models to enable major improvements. Having said that, Lebanon offers a potential avenue for AI integration into different domains especially healthcare services. It is crucial therefore to systematically document and share information on AI's role in clinical practice, enabling healthcare providers to acquire the knowledge and tools essential for its effective implementation in patient care. This review article delves into the prevalent trends shaping the integration of AI into various medical applications, taking as example the publications of different Lebanese universities between 2020 and 2024 to examine its potential uses, benefits and challenges, while also offering insights into its future development.

Thus, this study seeks to answer the following research questions:

1. What advances has AI brought to the healthcare sector in Lebanon?

2. What are the characteristics of the most recent Lebanese healthcare publications applying AI?

3. What AI methodologies have been applied for healthcare system in Lebanon?

4. What are the challenges faced by AI applications in health sciences field in Lebanon?

## 2. Materials and Methods

### 2.1. Selection Criteria

This study focuses on publications related to the application of artificial intelligence in the health and medical sciences field. The literature was sourced through searches in three major databases: Scopus, Web of Science (WoS), and PubMed. The search was limited to articles and reviews, excluding other document types such as conference proceedings, notes, abstracts, short communications, and letters to the editor. Only publications from the period between January 2020 and April 2024 were considered for inclusion. The selection criteria were refined further by using specific keywords related to artificial intelligence, as follows:

• "artificial intelligence" [Title/Abstract] OR "artificial intelligence" [MeSH terms]

• "machine learning" [Title/Abstract] OR "machine learning" [MeSH terms]

• "deep learning" [Title/Abstract] OR "deep learning" [MeSH terms]

• "robot" [Title/Abstract] OR "robot" [MeSH terms]

These keywords were combined with the keywords "health" OR "medic" (in either title or abstract) to ensure the relevance of the studies to the health and/or medical domains. Furthermore, publications with affiliations containing the term "Lebanon" or "LEBANON" were specifically targeted to focus on studies related to this geographic region. The studies reviewed were required to be published in English-language journals or conference proceedings. Non-English language publications were excluded to maintain consistency in language and interpretation. Only studies that explicitly mentioned the relevant terms in their titles, abstracts, or MeSH headings were included.

Because numerous observational studies were included, a scoping review was chosen over a systematic review because it allows for a broader exploration of the literature and identification of key concepts and research gaps, which is more suitable for this study given the heterogeneity of the included studies.

### 2.2. Data screening and assessment

A data extraction template was developed to extract all retrieved data. We identified the authors' affiliations from the fields of "affiliations". International collaboration was deemed to exist in an article if any author's affiliation was located outside

Lebanon. Citations of the articles in the WOS, SCOPUS and PUBMED databases were extracted in January 2024. To ensure eligibility, extracted results were reviewed by individual reviewers to ensure that the title/abstract includes any aspect of artificial intelligence. The review was divided into a set of phases or steps. The phases followed were: (1) review of previous publications, (2) definition of the inclusion and exclusion criteria, (3) definition of the search strategy, (4) definition of the quality criteria, (5) data extraction, (6) results, and (7) data analysis and report writing.

Extracted data was created in Microsoft excel for article screening, removing duplicate entries, and making appropriate corrections. In a two-phase procedure, two researchers conducted an independent reading of the titles and abstracts of the available articles, and subsequently examined full-text manuscripts to determine eligibility. To ensure methodological rigidity, a third auditor was consulted in case of any discrepancies, and a consensus on article eligibility was reached through rechecking the information. Additionally, a data extraction template for each article that underwent screening for inclusion was filled out. This template included fields for: (1) article title, (2) author names, (3) author count, (4) citations count, (5) journal name, (6) study design, (7) field of study, (8) authors characteristics such as gender, work status, tenure-track and faculty affiliation, (9) type of the collaboration whether institutional, national (co-authors don't belong to the same institution but reside in the same country) or international, and (10) Scimago Journal Rank (SJR) and quartile (Q) of the journal where Q1 comprises the quarter of the journals with the highest values, Q2 the second highest values, Q3 the third highest values and Q4 the lowest values.

Finally, the authors evaluated the methodological quality of the selected articles and classified them based on the type of AI used, and the purpose for which it was implemented. The composition of the review adhered to the PRISMA guidelines designed for scoping reviews [32].

## 2.3. Quality assessment

The quality assessment of the included studies was conducted using the Grading of Recommendations, Assessment, Development, and Evaluation (GRADE) tool [33]. Each potential source of bias was evaluated and classified into one of five categories: very low, low, intermediate, high, or unclear. The classification was determined based on the study design, with randomized trials receiving the highest confidence, while observational studies could be downgraded or upgraded accordingly. Case series and case reports were generally categorized as very low in quality.

## 2.4. Data Abstraction and synthesis

Main findings from the studies were reviewed using descriptive and analytical methods based on different variables outcomes (either publications characteristics such as author's count, collaboration, study design; journal quartile and citation counts or publications' content such as AI methodologies, benefits and challenges). Statistical analysis was performed using SPSS version 29 (IBM SPSS Software, Chicago, IL, USA). Continuous measures were summarized using either means and standard deviations or medians and interquartile ranges, depending on appropriateness. Categorical measures were summarized using frequencies and percentages. To compare the number of citations among different medical fields, ANOVA test was used. A *p-value* ≤0.05 was considered to be statistically significant.

## 2.5. Patient and Public involvement

Since this review did not involve human participants, there was no requirement to seek informed consent

## 3. Results

### 3.1 Study selection

A total of 2056 publications were identified through initial searches out of which 1340 studies focused on medical or health sciences. A first selection process involved the removal of studies not related to AI. Following an initial screening of titles

and abstracts, 270 contributions were selected for full-text review. A subsequent selection process was conducted to eliminate duplicates, letters to the editor, errata, and corrections to previous publications. After thoroughly reviewing the full articles, an additional 202 were deemed irrelevant and excluded, leaving a final set of 68 articles directly related to the research question. The most common reasons for excluding articles after full-text review included lack of relevance to the research question and AI not being the primary focus or methodology. A PRISMA flow-diagram is presented in Fig. 1 to illustrate the study selection process.

### 3.2. Publications' characteristics and quality assessment

A total of 68 publications related to the application of AI in the medical or healthcare field indexed in Scopus, WoS, and PubMed databases between January 2020 and April 2024 were included in our review. Fig. 2 displays the percentage of publications per year. The number of publications showed a steady rise from 2020 to 2023, peaking at its highest point (44.1%) in 2023 and an exponential increase in number is expected in 2024. Most of the publications were the results of a collaborative work with international universities (92.1%). Most of the authors were affiliated to the Lebanese American University (52.9%), followed by the American University of Beirut (25.0%).

The vast majority of all publications were modeling studies (41.2%) or descriptive/narrative reviews (29.4%). Among all publications, the most abundant health-related subject of interest was related to cancer diagnosis and management (25%) followed by cardiology (20.6%), then mental health diagnosis (10.3%) (Fig.3).

All the studies were characterized by multiple authorships. The average number of authors per study was $6.8 \pm 2.9$. More details about the characteristics of publications are summarized in Table 1. The GRADE evaluation framework was

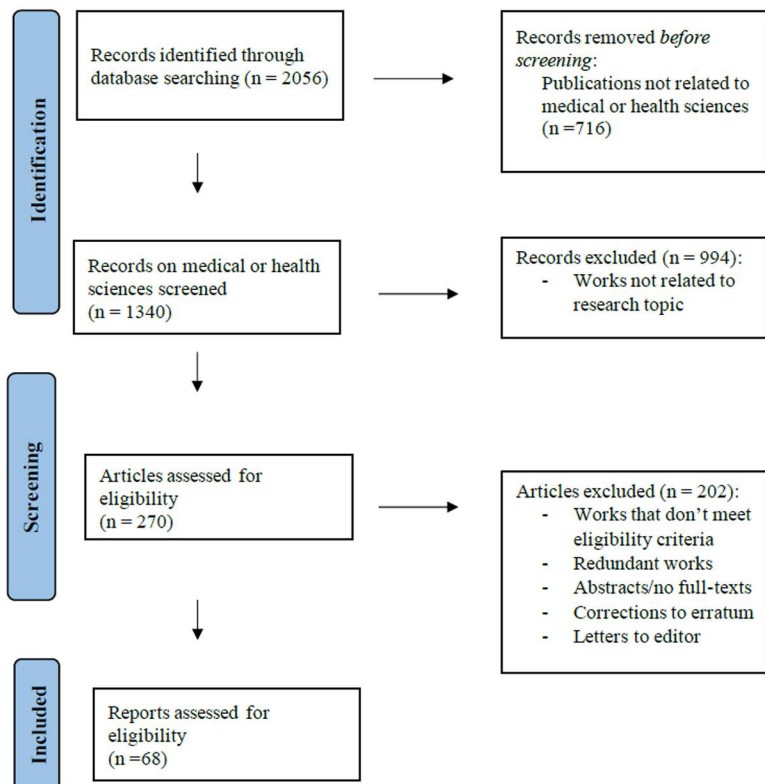

**Fig 1. PRISMA flow diagram of studies selection process.** (Caption: Process of inclusion and exclusion of studies).

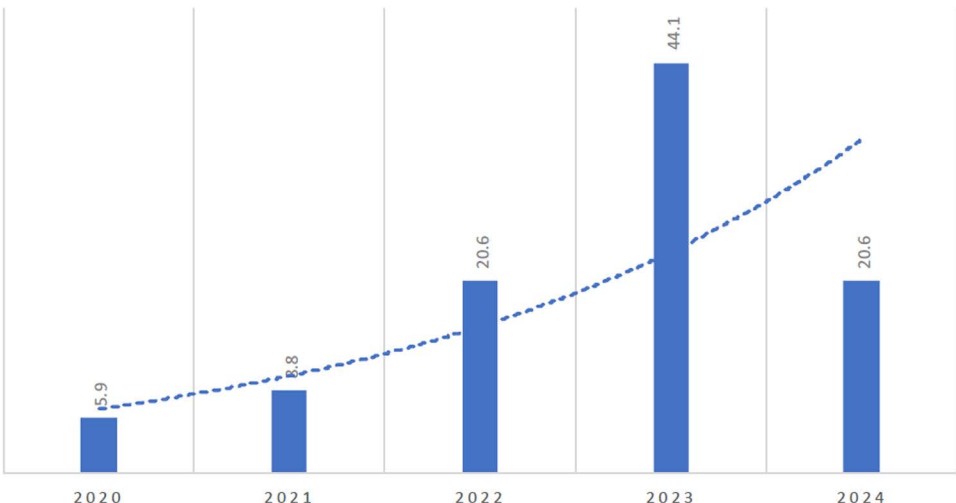

**Fig 2. Percentage of AI in health studies across the years.** (Caption: Percentages of health studies that apply AI across the years).

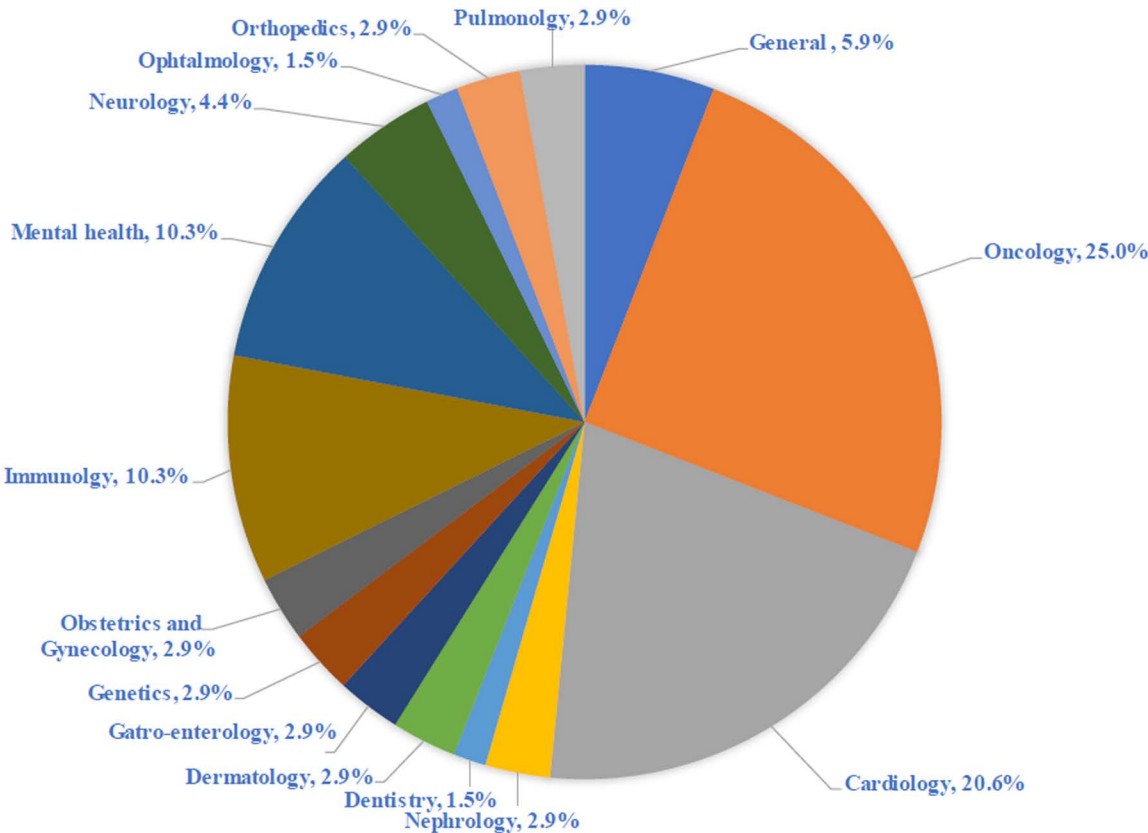

**Fig 3. Main areas of AI application.** (Caption: percentage of health-related subject that apply AI).

**Table 1.  Characteristics of the publications.**

|  | Number (N) | Percentage (%) |
|---|---|---|
| **Authors' count** | | |
| < 2 | 4 | 5.9 |
| ≥ 2 | 64 | 94.1 |
| **Work Collaboration** | | |
| Local | 3 | 4.4 |
| National | 3 | 4.4 |
| International | 62 | 91.2 |
| **Study Design** | | |
| Invitro/in-vivo/in-silico | 7 | 10.3 |
| Cohort | 3 | 4.4 |
| Cross-sectional | 5 | 7.4 |
| Modeling | 28 | 41.2 |
| Review (general) | 20 | 29.4 |
| Systematic review and Meta-analysis | 5 | 7.4 |
| **Academic Institution** | | |
| American University of Beirut | 17 | 25 |
| Lebanese American University | 36 | 52.9 |
| Lebanese University | 5 | 7.4 |
| Beirut Arab University | 3 | 4.4 |
| Saint Joseph University | 7 | 10.3 |
| Others | 3 | 4.4 |
| Studies per database | | |
| Pubmed | 62 | 91.1 |
| Scopus and WoS | 6 | 8.9 |

utilized to assess the primary outcomes of the included studies. Interventional studies were classified as having high-quality evidence, while observational studies were rated as moderate to low in quality. Literature reviews, however, were considered to provide a very low level of evidence quality.

### 3.3. Characteristics of the journals and citation numbers

As detailed in Table 2, all of the publications authored by all scholars appeared in peer-reviewed journals. The majority of the publications were published in SJR Q1 journals (66.2%) while 20 papers (29.28%) belonged to Q2 journals. Only three publications were submitted to journals of lower SJR rank. The most repetitive journal is Scientific Reports followed by Diagnostics.

The majority of the studies were cited at least once (73.3%) while only 26.7% received no citations. All publications were cited a total of 454 times, the mean citation number was $7.57 \pm 12.31$, and 40% of the publications were cited 2–10 times. As shown in Fig. 4, immunology field received the highest number of citations (mean±SD: $11.17 \pm 11.8$) followed by cardiology (mean±SD: $8.75 \pm 16.36$). However, this difference in the number of citations did not reach statistical significance ($p = 0.092$).

### 3.4. Framework of Research Classification

All included studies were organized into three dimensions: AI methodologies, benefits, and challenges. All articles (100%) were attributed an AI methodology and 100% of them were aggregated under the benefits dimension. Seven studies (10.3%) were aggregated under the challenges dimension.

**Table 2. Characteristics of the journals.**

|  | Number (N) | Percentage (%) |
|---|---|---|
| **Journals per quartile** | | |
| Q1 | 45 | 66.2 |
| Q2 | 20 | 29.2 |
| Q3 | 3 | 4.4 |
| **Citations** | | |
| 0 | 16 | 26.7 |
| 1 | 8 | 13.3 |
| 2-10 | 24 | 40.0 |
| >10 | 12 | 20.0 |
| **Most repetitive journals** | | |
| Scientific Reports | 9 | 13.2 |
| Diagnostics | 4 | 5.9 |
| Sensors | 2 | 2.9 |
| Journal of Vascular Surgery | 2 | 2.9 |
| Studies in Health Technology and Informatics | 2 | 2.9 |

*Q: Journal Quartile

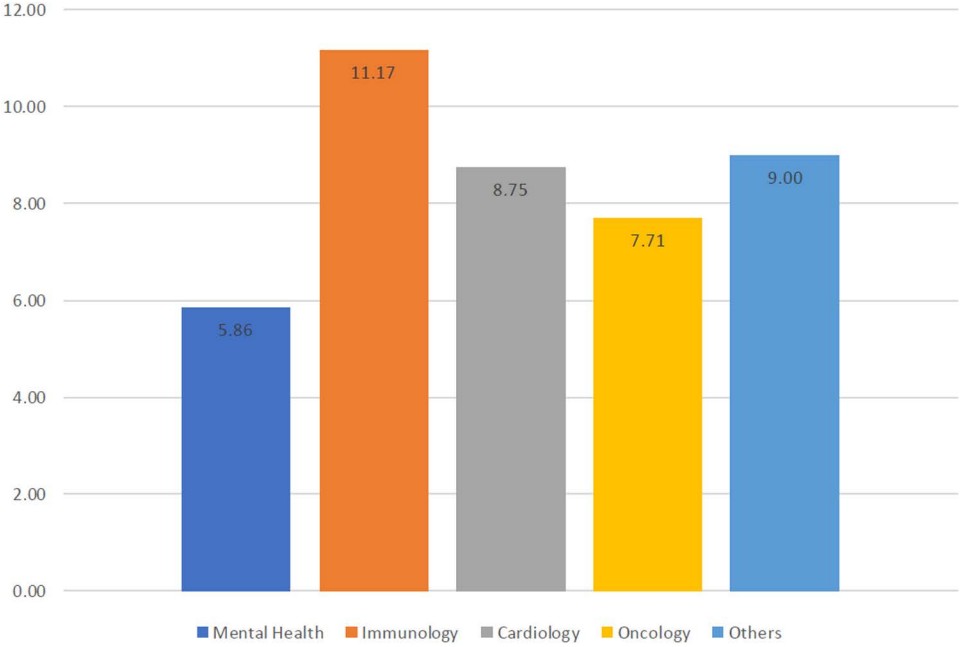

**Fig 4. Differences in the average number of citations among different medical fields.** (Caption: Mean citation numbers across different fields of study).

Table 3 shows the classification framework related to AI areas/methodologies that were discussed in the selected studies and contains specific categories for this dimension, such as data or multimedia. In addition, in each category, relevant factors were identified that defined the category. Table 4 illustrates the classification framework related to both AI enabled

healthcare benefits and challenges associated with the use of AI in different healthcare fields and involves specific categories for this dimension, such as challenges related to data integration and patient safety.

**3.4.1. AI Methodologies.** In the context of applying AI in healthcare, methodology refers to the procedures or areas healthcare institutions implement to utilize AI. The two main AI areas are related to either data processing or multimedia processing. The data processing includes both Deep Learning Methods (DLM) or Machine Learning (ML); and the multimedia processing can be divided into AI devices/imaging or video processing or virtual reality [34].

**A. Data processing** Of the selected studies, 75% (n = 51) applied AI technology based on data processing models, of which 66% (n = 34) applied the DLM and the other 34% were based on ML. The oncology field was the primary therapeutic area utilizing deep learning models (DLM) as the main AI tool (n = 11). Among these studies, four focused on brain tumors [35–38], while others explored its application in various cancers, including prostate [39], skin [40], breast [41], bone [42], pancreatic [43], and colon cancer [44], as well as in the broader context of cancer metastasis prediction [45]. Additionally, six cardiology studies examined the use of DLM for diagnosing and detecting various cardiovascular diseases [46–51], followed by five studies in infectious diseases [52–56], mental health [57,58], gastroenterology [59], neurology [60,61], and several other medical fields [62–68]. As for AI-based ML, it was most commonly mentioned in studies from the cardiology unit (n = 6) [69–74] followed by mental studies [75–77] then cancer studies [78,79] in addition to diverse other domains [80–85]. More details on AI data processing tools can be found in Table 3.

**B. Multimedia processing** As shown in Table 3, 25% of the included studies applied AI technology based on multimedia processing models, of which 53% (n = 9) discussed the use of AI devices, imaging or video processing, and 47% applied virtual reality to improve diagnosis or detection of certain diseases. Studies from the oncology field (n = 3) predominantly used AI devices to detect and diagnose some types of cancers such as brain [86], gastric [87] or colon [85]. Other fields that utilized the same AI tool included gastroenterology, where it was used to assess the role of nanotechnology in the formulation of nutraceuticals [89], pulmonary chest radiography [90], the detection and classification of infectious diseases [91], kidney diseases [92] and the prediction, detection, and classification of cardiovascular conditions such as atrial fibrillation [93,94].. Virtual reality played an important role in mental health studies particularly in describing digital mental health procedures and models for detection of depression [95,96]. Moreover, it was also used as an emerging application in dentistry [97], gynecological [98], orthopedic procedures [99] and cancer detection [100] in addition to infectious disease [101] as well as general medical research [102].

**3.4.2. Benefits of AI.** This aspect refers to the achievable benefits gained through AI utilization. These include benefits to individuals, such as automated decision-making, patient monitoring and prognosis, early diagnosis, and process simplification and therapeutic management. Additionally, they include benefits to organizations like optimizing workflow management, improving performance, and ensuring data availability.

**A. Benefits to individuals** The analysis of the included studies revealed a growing interest in exploring the potential of AI-based support systems for enhancing early disease diagnosis. This is evident from the fact that a significant number of the studies (n = 14) focused on this particular aspect. Of these, the most commonly repetitive therapeutic field was related to cancer [39,42,43,78] followed by mental health (depression and anxiety) [75–77] and other infectious diseases such as tuberculosis [91], crop disease [56] and coronavirus detection [47]. Moreover, a very recent study conducted by Ghazi et al. investigated the use of AI machines to predict acute kidney injury [80].

Furthermore, the use of AI-powered simulations to enhance healthcare decision-making abilities were discussed in different studies across diverse therapeutic domains including gynecology [82], cardiology [70] and cognitive health [57]. Similarly, other studies concluded that AI-based resources have the potential to improve patient outcomes and prognosis in various oncological, cardiovascular and immunological studies [36,45,54,70,72,79]. Likewise, another AI benefit - process simplification - has been proven in 14.7% (n = 10) of the studies. This implication was seen in different medical domains including mainly cardiology imaging [46,71,73,74,94] and physiological measurements of mental health diseases [58,96]. In the same manner, AI-assisted tools, such as 3D printing models and theragnostic applications, have shown

**Table 3. AI methodologies applied in the healthcare setting.**

| Study | Title | Reference | Date | Medical area | AI Category | AI Area | Summarized content |
|---|---|---|---|---|---|---|---|
| Gumaei et al. | Feature selection with ensemble learning for prostate cancer diagnosis from microarray gene expression | 39 | 2021 | Oncology | Data processing | Deep learning methods/ modeling | Use of microarray gene expression for prostate cancer diagnosis |
| Zafar et al. | Detection Based on Machine/Deep Learning Techniques: A Comprehensive Survey | 40 | 2023 | | | | Use of deep learning techniques for skin lesion analysis and cancer detection |
| Kumar et al. | Brain tumor classification using deep neural network and transfer learning | 35 | 2023 | | | | Use of deep neural network and transfer learning for brain tumor classification |
| Ghanem et al. | Deep Learning Approaches for Glioblastoma Progno-sis in Resource-Limited Settings: A Study Using Basic Patient Demographic, Clinical, and Surgical Inputs | 36 | 2023 | | | | Use of deep learning approaches for glioblastoma prognosis |
| Ullah et al. | BrainNet: a fusion assisted novel optimal framework of residual blocks and stacked autoencoders for multimodal brain tumor classification | 38 | 2024 | | | | Use of novel optimal framework of residual blocks and stacked autoencoders for multimodal brain tumor classification |
| Zaylaa et al. | Advancing Breast Cancer Diagnosis through Breast Mass Images, Machine Learning, and Regression Models | 41 | 2024 | | | | Use of breast mass images, machine learning, and regression models for breast cancer diagnosis |
| Dhasmana et al. | Integrative big transcriptomics data analysis implicates crucial role of MUC13 in pancreatic cancer | 43 | 2023 | | | | Use of integrative big transcriptom-ics data analysis in the detection of MUC13 in pancreatic cancer |
| Yagin et al. | Cancer Metastasis Prediction and Genomic Biomarker Identification through Machine Learning and eXplainable Artificial Intelligence in Breast Cancer Research | 45 | 2023 | | | | Use of explainable artificial intelligence in cancer metastasis prediction |
| Halabi et al. | Unveiling a Biomarker Signature of Meningioma: The Need for a Panel of Genomic, Epigenetic, Proteomic, and RNA Biomarkers to Advance Diagnosis and Prognosis | 37 | 2023 | | | | Use of AI in unveiling RNA biomarkers for the diagnosis and prognosis of meningioma |
| Rajinikanth et al. | Colon histology slide classification with deep-learning framework using individual and fused features | 44 | 2023 | | | | Use of deep-learning framework for colon histology classification |
| Magdy et al. | Bone metastasis detection method based on improving golden jackal optimization using whale optimization algorithm | 42 | 2023 | | | | Bone metastasis detection using ai optimization algorithm |
| Rammal et al. | Machine learning techniques on homological persistence features for prostate cancer diagnosis | 78 | 2022 | | | Machine Learning | Use of machine learning techniques for prostate cancer diagnosis |
| Ghaith et al. | Using machine learning to predict 30-day readmission and reoperation following resection of supratentorial high-grade gliomas: an ACS NSQIP study involving 9418 patients | 79 | 2023 | | | | Use of machine learning to predict prognosis associated with glioma |

*(Continued)*

**Table 3.** (Continued)

| Study | Title | Reference | Date | Medical area | AI Category | AI Area | Summarized content |
|---|---|---|---|---|---|---|---|
| Lucchetti et al. | Smart nano-sized extracellular vesicles for cancer therapy: Potential theranostic applications in gastrointestinal tumors | 87 | 2023 | | Multimedia processing | AI devices/ Imaging processing | Application of smart nano-sized extracellular vesicles for gastrointestinal cancer therapy |
| Hage Chehade et al. | Lung and colon cancer classification using medical imaging: a feature engineering approach | 88 | 2022 | | | | Use of ai medical imaging for lung and colon cancer classification |
| Felefly et al. | An Explainable MRI-Radiomic Quantum Neural Network to Differentiate Between Large Brain Metastases and High-Grade Glioma Using Quantum Annealing for Feature Selection | 86 | 2023 | | | | MRI-radiomic quantum neural network to differentiate between large brain metastases and high-grade glioma |
| Atat et al. | 3D modeling in cancer studies | 100 | 2022 | | | Virtual reality | Use of 3d modeling in cancer studies |
| Walsh et al. | A speckle-tracking strain-based artificial neural network model to differentiate cardiomyopathy type | 46 | 2020 | Cardiology | Data processing | Deep learning methods/ modeling | Description of a speckle-tracking strain-based artificial neural network model to differentiate cardiomyopathy type |
| Ahmad et al. | A comparison of artificial intelligence-based algorithms for the identification of patients with depressed right ventricular function from 2-dimentional echocardiography parameters and clinical features | 47 | 2020 | | | | Comparison of ai-based algorithms for the identification of depressed right ventricular function with 2-d echocardiography |
| Helwan et al. | Conventional and deep learning methods in heart rate estimation from RGB face videos | 48 | 2024 | | | | Description of conventional and deep learning methods in heart rate estimation from RGB face videos |
| Guldogan et al. | A proposed tree-based explainable artificial intelligence approach for the prediction of angina pectoris | 49 | 2023 | | | | Description of AI approach for the prediction of angina pectoris |
| Mitu et al. | A stroke prediction framework using explainable ensemble learning | 50 | 2024 | | | | Use of AI explainable ensemble learning to predict a stroke |
| Li et al. | Development and evaluation of a prediction model for peripheral artery disease-related major adverse limb events using novel biomarker data | 51 | 2024 | | | | Development and evaluation of a prediction model for peripheral artery disease |
| Barakett-Hamade et al. | Is Machine Learning-derived Low-Density Lipoprotein Cholesterol estimation more reliable than standard closed form equations? Insights from a laboratory database by comparison with a direct homogeneous assay | 69 | 2021 | | | Machine Learning | Comparison of machine learning-derived low-density lipoprotein cholesterol to standard closed form equations |
| Li et al. | Machine learning to predict outcomes following endovascular abdominal aortic aneurysm repair | 72 | 2024 | | | | Use of machine learning to predict outcomes of endovascular abdominal aortic aneurysm repair |
| Li et al. | Predicting outcomes following open revascularization for aortoiliac occlusive disease using machine learning | 70 | 2023 | | | | Use of machine learning to predict outcomes of open revascularization for aortoiliac occlusive disease |
| Li et al. | Predicting Major Adverse Cardiovascular Events Following Carotid Endarterectomy Using Machine Learning | 71 | 2023 | | | | Use of machine learning to predict major adverse cardiovascular events following carotid endarterectomy |
| Hammoud et al. | Predicting incomplete occlusion of intracranial aneurysms treated with flow diverters using machine learning models | 74 | 2023 | | | | Use of machine learning to predict incomplete occlusion of intracranial aneurysms treated with flow diverters |
| Li et al. | Predicting outcomes following lower extremity open revascularization using machine learning | 73 | 2024 | | | | Use of machine learning to predict outcomes following lower extremity open revascularization |

*(Continued)*

**Table 3.** (Continued)

| Study | Title | Reference | Date | Medical area | AI Category | AI Area | Summarized content |
|---|---|---|---|---|---|---|---|
| Serhal et al. | Overview on prediction, detection, and classification of atrial fibrillation using wavelets and AI on ECG | 93 | 2022 | | Multi-media process-ing | AI devices/ Imaging processing | Overview on prediction, detection, and classification of atrial fibrillation using wavelets and AI on electrocardiogram |
| Moshawrab et al. | Smart Wearables for the Detection of Cardiovascular Diseases: A Systematic Literature Review | 94 | 2023 | | | AI devices/ video processing | Use of smart wearables for the detection of cardiovascular diseases |
| Ghazi et al. | Biomarkers vs Machines: The Race to Predict Acute Kidney Injury | 80 | 2024 | Nephrol-ogy | Data pro-cessing | Machine Learning | Comparison of biomarkers to machines in predicting acute kidney injury |
| Alnazer et al. | Recent advances in medical image processing for the evaluation of chronic kidney disease | 92 | 2021 | | Multi-media process-ing | AI devices/ Imaging processing | Medical image processing for the evaluation of chronic kidney disease |
| Helwan et al | Radiologists versus Deep Convolutional Neural Networks: A Comparative Study for Diagnosing COVID-19 | 52 | 2021 | Infection | Data pro-cessing | Deep learning methods/ modeling | Comparison of radiology to deep convolutional neural network in diagnosing COVID-19 |
| Rashid et al. | White blood cell image analysis for infection detection based on virtual hexagonal trellis (VHT) by using deep learning | 53 | 2023 | | | | Use of using deep learning method for white blood cell image analysis and infection detection |
| Tarek et al. | An Optimized Model Based on Deep Learning and Gated Recurrent Unit for COVID-19 Death Prediction | 54 | 2023 | | | | Use of using deep learning method for covid-19 death prediction |
| Amin et al. | Microscopic parasite malaria classification using best feature selection based on generalized normal distribution optimization | 55 | 2024 | | | | Use of generalized normal distribution optimization for malaria classification |
| Ngugi et al. | Revolutionizing crop disease detection with computational deep learning: a comprehensive review | 56 | 2024 | | | | Use of computational deep learning for crop disease detection |
| Saleh et al. | A three-dimensional A549 cell culture model to study respiratory syncytial virus infections | 101 | 2020 | | Multi-media process-ing | Virtual reality | Description of 3D model to study respiratory syncytial virus infections |
| Acharya et al. | AI-Assisted Tuberculosis Detection and Classification from Chest X-Rays Using a Deep Learning Normalization-Free Network Model | 91 | 2022 | | | AI devices/ Imaging processing | Use of a deep learning normalization-free network model for tuberculosis detection and classification |

*(Continued)*

**Table 3.** (Continued)

| Study | Title | Reference | Date | Medical area | AI Category | AI Area | Summarized content |
|---|---|---|---|---|---|---|---|
| Javed et al. | Toward explainable AI-empowered cognitive health assessment | 57 | 2023 | Mental health | Data processing | Deep learning methods/modeling | Description of AI-empowered cognitive health assessment |
| Jaber et al. | Medically-oriented design for explainable AI for stress prediction from physiological measurements | 58 | 2022 | | | | Use of AI model based on physiological measurements to predict stress occurrence |
| Qasrawi et al. | Machine learning techniques for predicting depression and anxiety in pregnant and postpartum women during the COVID-19 pandemic: a cross-sectional regional study | 75 | 2022 | | | Machine Learning | Use of machine learning techniques for the prediction of depression and anxiety in pregnant and postpartum women during the COVID-19 |
| Mahalingam et al. | A Machine Learning Study to Predict Anxiety on Campuses in Lebanon | 76 | 2023 | | | | Use of machine learning to predict anxiety |
| El Morr et al. | Predictive Machine Learning Models for Assessing Lebanese University Students' Depression, Anxiety, and Stress During COVID-19 | 77 | 2024 | | | | Use of machine learning models to assess Lebanese university students' depression, anxiety, and stress during covid-19 |
| Boulos et al. | An Iterative and Collaborative End-to-End Methodology Applied to Digital Mental Health | 95 | 2021 | | Multimedia processing | Virtual reality | Description of digital mental health |
| Kabbara et al. | An electroencephalography connectome predictive model of major depressive disorder severity | 96 | 2022 | | | | Description of electroencephalogram predictive model for detecting the severity of major depressive disorder |
| Ghanem et al. | Limitations in Evaluating Machine Learning Models for Imbalanced Binary Outcome Classification in Spine Surgery: A Systematic Review | 81 | 2023 | Orthopedics | Data processing | Machine Learning | Description of the limitations of machine learning models used in spine surgery |
| Yammine et al. | Clinical outcomes of the use of 3D printing models in fracture management: a meta-analysis of randomized studies | 99 | 2022 | | Multimedia processing | Virtual reality | Description of the outcomes of the use 3d printing models in fracture management |
| Ramzan et al. | Gastrointestinal tract disorders classification using ensemble of InceptionNet and proposed GITNet based deep feature with ant colony optimization | 59 | 2023 | Gastroenterology | Data processing | Deep learning methods/modeling | Use of deep learning models for the classification of gastrointestinal tract disorders |
| Dangi et al. | Nanotechnology impacting probiotics and prebiotics: a paradigm shift in nutraceuticals technology | 89 | 2023 | | Multimedia processing | AI devices/Imaging processing | Use of nanotechnology impacting probiotics and prebiotics in the composition of nutraceuticals |
| Hammoud et al. | Can machine learning models predict maternal and newborn healthcare providers' perception of safety during the COVID-19 pandemic? A cross-sectional study of a global online survey | 82 | 2022 | Gynecology | Data processing | Machine Learning | Survey of the use of machine learning models to predict maternal and newborn healthcare providers' perception of safety during the covid-19 pandemic |
| Jerbaka et al. | Outcomes of robotic and laparoscopic surgery for benign gynaecological disease: a systematic review | 98 | 2022 | | Multimedia processing | Virtual reality | Description of the outcomes of robotic and laparoscopic surgery for benign gynecological disease |

*(Continued)*

**Table 3.** (Continued)

| Study | Title | Reference | Date | Medical area | AI Category | AI Area | Summarized content |
|---|---|---|---|---|---|---|---|
| Hallal et al. | TempoMAGE: a deep learning framework that exploits the causal dependency between time-series data to predict histone marks in open chromatin regions at time-points with missing ChIP-seq datasets | 62 | 2021 | Genetics | Data processing | **Deep learning methods/ modeling** | Use of a deep learning framework to predict histone marks in open chromatin regions at time-points with missing chip-seq datasets |
| de Brevern et al. | Current status of PTMs structural databases: applications, limitations and prospects | 63 | 2022 | | | | Description of the current status of post-translational modifications (PTMs) structural databases |
| Ali et al. | Parkinson's disease detection based on features refinement through L1 regularized SVM and deep neural network | 60 | 2024 | Neurology | Data processing | **Deep learning methods/ modeling** | Use of deep neural network for detection of Parkinson's disease |
| Voigtlaender et al. | Artificial intelligence in neurology: opportunities, challenges, and policy implications | 61 | 2024 | | | | General description of AI implications in neurology |
| Chedid et al. | The development of an automated machine learning pipeline for the detection of Alzheimer's Disease | 83 | 2022 | | | **Machine Learning** | Description of the development of an automated machine learning pipeline for the detection of Alzheimer's disease |
| Hussain et al. | SkinNet-INIO: Multiclass Skin Lesion Localization and Classification Using Fusion-Assisted Deep Neural Networks and Improved Nature-Inspired Optimization Algorithm | 64 | 2023 | Dermatology | Data processing | **Deep learning methods/ modeling** | Description of the use of deep neural networks and algorithms for multiclass skin lesion localization and classification |
| Bibi et al. | MSRNet: Multiclass Skin Lesion Recognition Using Additional Residual Block Based Fine-Tuned Deep Models Information Fusion and Best Feature Selection | 65 | 2023 | | | | Description of the use of fine-tuned deep models' information fusion and best feature selection for multiclass skin lesion |
| Al-Sheikh et al. | Multi-class deep learning architecture for classifying lung diseases from chest X-Ray and CT images | 66 | 2023 | Pneumology | Data processing | **Deep learning methods/ modeling** | Use of deep learning architecture for classifying lung diseases |
| Dasegowda et al. | Suboptimal Chest Radiography and Artificial Intelligence: The Problem and the Solution | 90 | 2023 | | **Multi-media processing** | **AI devices/ Imaging processing** | Use of AI in the advancement of suboptimal chest radiography |

*(Continued)*

**Table 3.** (Continued)

| Study | Title | Reference | Date | Medical area | AI Category | AI Area | Summarized content |
|---|---|---|---|---|---|---|---|
| Askin et al. | Artificial Intelligence Applied to clinical trials: opportunities and challenges | 67 | 2023 | **General** | Data processing | **Deep learning methods/ modeling** | Artificial intelligence applied to clinical trials: opportunities and challenges |
| Saab et al. | Early Prediction of All-Cause Clinical Deterioration in General Wards Patients: Development and Validation of a Biomarker-Based Machine Learning Model Derived From Rapid Response Team Activations | 84 | 2022 | | | **Machine Learning** | Development and validation of a biomarker-based machine learning model to predict all-cause clinical deterioration in general wards |
| Saab et al. | .Comparison of Machine Learning Algorithms for Classifying Adverse-Event Related 30-Day Hospital Readmissions: Potential Implications for Patient Safety | 85 | 2020 | | | | Use of different machine learning algorithms for classifying adverse-event related 30-day hospital readmissions and patient safety |
| Malik et al. | Emerging Applications of Nanotechnology in Healthcare and Medicine | 102 | 2023 | | **Multimedia processing** | **Virtual reality** | Description of emerging applications of nanotechnology in healthcare and medicine |
| Malik et al. | Emerging Applications of Nanotechnology in Dentistry | 97 | 2023 | **Dentistry** | **Multimedia processing** | **Virtual reality** | Emerging applications of nanotechnology in dentistry |
| Venkatapathappa et al. | Ocular Pathology and Genetics: Transformative Role of Artificial Intelligence (AI) in Anterior Segment Diseases | 68 | 2024 | **Opthtalmology** | Data processing | **Deep learning methods/ artificial intelligence** | Description and use of transformative AI in ocular genetics |

to offer benefits in therapeutic management and clinical outcomes, as evidenced by studies conducted by Yammin et al. (2022) [87] and Lucchetti et al (2024) [99].

**B. Benefits to organizations**   Organizations use AI applications and tools to provide a better workflow management [89,95], overcome data availability and integration [61,63,67,68,81,90,93,97,100,102] and improve organizational improvement of different medical departments such as cancer imaging and diagnosis [35,37,38,40,41,44,88], cardiovascular disease prediction and management [47,49,69], dermatological lesions recognition [64,65], or infection detection [53,101]Moreover, in his prior studies, Saab et al. described the use of ML for general therapeutic purposes such as the early prediction of all-cause clinical deterioration [84] and the classification of adverse-event related 30-day hospital readmissions [85]. More details on the benefits of AI applications are summarized in Table 4.

**3.4.3. Challenges of AI.**  As detailed in Table 4, a number of challenges may deter organizations from using AI. Nevertheless, the number of studies focusing on AI challenges has been the lowest in the past ten years. The most occurring challenges in this review were related to data integration and/or patient safety.

Challenges related to data integration consist of data availability in cardiology and immunology units [50,56] as well as data digitization and consolidation in genetics and medical research field [63,67]. AI Challenges, from the patient's perspective, are related to decision errors [102] and data errors in both neurologic and orthopedic surgeries [61,81].

## 4. Discussion

### 4.1. Principal findings

AI is gaining significant attention across various fields, including medicine. The purpose of this review was to gather and summarize the existing information regarding the use of artificial intelligence in the healthcare setting in Lebanon. AI models are primarily theoretical, utilizing automation or optimization technologies. They are predominantly designed for clinical care and diagnostic applications, with the majority aimed at supporting human decision-making. However, many of these models remain conceptual and lack empirical evidence to support their effectiveness [103]. Most of the studies have been recently published (between 2020 and 2024) and are characterized by their prediction/modeling design which is expected given the timeline and the future perspective of AI in healthcare research. To add, many papers have multiple authors, with international collaboration being the most prevalent form of authorship. This is essential due to the topic's significance, requiring expertise from various domains given its novelty. This is also consistent with other prior publications, confirming that research collaboration adds benefits for both the researchers and the organizations and enhances the quality of research resulting in higher numbers of scholarly output [104].

The majority of the analyzed studies are published in top-tier (Q1) journals specializing in healthcare, medical information systems, informatics, and machine learning, indicating a strong focus on AI applications in these areas. These publications have also gained multiple citations, demonstrating their significant impact across various research fields, irrespective of the specific therapeutic domain. After a thorough review of the pertinent studies, our results have summarized the main methodologies of the AI implied in the healthcare setting as well as the benefits and challenges associated with its use. The effectiveness of AI, in medical care, is influenced by the type of intelligence utilized and its applications [105,106]. Accordingly, this review, similar to other previous studies, found that ML models are commonly used in diagnostic support systems [107,108]. The popularity of this AI type is mainly due to its efficiency and cost-effectiveness in performing human tasks [109]. Similarly, our findings indicate that AI applications in healthcare are predominantly focused on diagnosing and predicting diseases such as cancer, cardiovascular diseases, infectious diseases, and neurological disorders. This focus enhances patient care by allowing healthcare professionals to spend more time with patients, adopt a holistic care approach, and improve patient satisfaction [110]. Moreover, the deep learning sequence is also considered as an essential improvement of different strategies utilized to upgrade healthcare practice [111]. With the use of modern computational methods and computer learning, more data would become available, which could give insights into many different medical and healthcare practices [112]. Our study has shown that AI makes it easier to turn data into concrete

**Table 4. AI enabled benefits and challenges in the healthcare setting.**

| Dimension | Category | Type | Studies |
|---|---|---|---|
| Benefits | **Individual** | **Disease diagnosis** | Rammal et al., 2022 [78]; Dhasmana et al., 2023 [43]; Ahmad et al., 2020 [47]; Acharya et al., 2022 [91]; Qasrawi et al., 2022 [75]; Mahalingam et al., 2023 [76]; El Morr et al., 2024 [77]; Chedid et al., 2022 [83]; Mitu et al., 2024 [50]; Ghazi et al., 2024 [80]; Helwan et al., 2021 [52]; Ngugi et al., 2024 [56]; Gumaei et al., 2021 [39]; Magdy et al., 2023 [42] |
| | | **Patient monitoring/ prognosis** | Ghanem et al., 2023 [36]; Ghaith et al., 2023 [79]; Tarek et al., 2023 [54]; Yagin et al., 2024 [45]; Li et al., 2024 [70]; Li et al., 2023 [72] |
| | | **Decision-making** | Javed et al., 2023 [57]; Yammine et al., 2022 [99]; Li et al., 2024 [51]; Hammoud et al., 2022 [82] |
| | | **Process simplification** | Walsh et al., 2020 [46]; Moshawrab et al., 2023 [94]; Li et al., 2023 [30]; Hammoud et al., 2023 [74]; Li et al., 2024 [71]; Jerbaka et al., 2022 [98]; Amin et al., 2024 [55]; Jaber et al., 2022 [85]; Kabbara et al., 2022 [96]; Felefly et al., 2023 [86] |
| | | **Therapeutic management** | Lucchetti et al., 2023 [89] |
| | **Organization** | **Performance improvement** | AlNazer et al., 2021 [92]; Hussain et al., 2023 [64]; Bibi et al., 2023 [65]; Ramzan et al., 2023 [59]; Hallal et al., 2021 [62]; Saleh et al., 2020 [101]; Rashid et al., 2023 [53]; Ali et al., 2024 [60]; Saab et al., 2020 [85]; Al-Sheikh et al., 2023 [66]; Saab et al., 2022 [84]; Hage Chehade et al., 2022 [88]; Zafar et al., 2023 [40]; Kumar et al., 2023 [35]; Halabi et al., 2023 [37]; Rajinikanth et al., 2023 [44]; Ullah et al., 2024 [38]; Zaylaa et al., 2024 [41]; Barakett-Hamade et al., 2021 [71]; Helwan et al., 2024 [48]; Guldogan et al., 2023 [49] |
| | | **Data availability** | Askin et al., 2023 [67]; Voigtlaender et al., 2024 [61]; Venkatapathappa et al., 2024 [68]; Dasegowda et al., 2023 [90]; Ghanem et al., 2023 [81]; Malik et al., 2023 [102]; Malik et al., 2023 [97]; Atat et al., 2022 [100]; Serhal et al., 2022 [93]; de Brevern et al., 2022 [63] |
| | | **Workflow management** | Dangi et al., 2023 [89]; Boulos et al., 2021 [95] |
| Challenges | **Data Integration** | **Data availability** | Mitu et al, 2024 [50]; Ngugi et al., 2024 [56] |
| | | **Digitalization** | Askin et al., 2023 [67]; deBrevern et al., 2022 [63] |
| | **Patient Safety** | **Data errors** | Voigtlaender et al., 2024 [61]; Ghanme et al., 2023 [81] |
| | | **Decision errors** | Malik et al., 2023 [102] |

and predictable observations to improve disease diagnosis, patient monitoring, decision-making, and deliver high-quality therapeutic management in many different fields such as oncology, cardiology, and even mental or neurologic diseases. Similar to previous research utilizing these computational tools, it has also been demonstrated that healthcare professionals can use data not only to describe current events but also to predict future outcomes and create opportunities that improve organizational performance and optimize workflow management [113,114]. On the other hand, other AI technologies, including robotic process automation and physical robots, have demonstrated their effectiveness in some specific studies [115]. These AI solutions enhance disease identification and management, as well as adherence to treatment protocols. However, based on the results of our review, it can be inferred that these AI types are less familiar in the Middle Eastern region compared to more prevalent technologies like DLP and ML.

In addition to the benefits on the organizational level, one of the notable benefits of AI techniques is the potential support for comprehensive health services management on the individual level. For instance, and as mentioned before, an AI system can offer health professionals continuous updates on medical issues to improve patient safety, and promote treatment efficacy [116,117]. Other AI tools permit the integration of patient information tools and the generation of outcome predictions [118,119]. AI is effective in analyzing large datasets and generating innovative, relevant solutions for health practitioners, enhancing patient care, diagnosis, and treatment options which usually require significant time and effort. By making automation easier and more available with minimal human intervention, AI can even surpass human performance in certain medical scenarios, such as radiology, cardiology, and tumor detection [120].

On the contrary of earlier research that emphasized AI's benefits in healthcare, such as enhanced prediction and decision-making, our study focuses on the challenges associated with AI implementation in medical settings. These include issues related to data availability, digitalization, and errors during data processing and decision-making. For example, Mitu et al. reported the challenges regarding data integration and availability [50] and Ghanem (b) et al. reported AI decision errors faced in orthopedic surgery [81]. Likewise, some authors highlighted these difficulties which hinder achieving clinically relevant results [121]. Sometimes the applied algorithm may be inappropriate for the data, or the data may still lack enough reliability for use in classification algorithms like neural networks and decision trees. Several previous studies have also demonstrated these AI -related challenges and possible decision-making problems in the health domain and discussed their solutions [122–124] which explains why major AI companies are actively identifying priority areas, opportunities, and recommendations to address these concerns in healthcare practice [125,126].

## 4.2. Limitations

Despite the results and findings obtained, the presented article has several limitations. Although most analyzed types demonstrated positive results regarding AI usage, a bigger number of studies is needed to address the current limitations of these systems and fulfill the requirements of professionals in AI-based system development. Moreover, the application of AI in healthcare, particularly in low socio-economic countries like Lebanon, is still developing and lacks substantial evidence. This might be due to the predominance of observational and descriptive studies, which exhibit a great heterogeneity in AI types and settings. Therefore, generalizations of the proposed results should be done cautiously. Additionally, this review did not include a meta-analysis or systematic review, as the significant heterogeneity in study methodologies and designs made it challenging to synthesize the findings in a standardized manner. Conducting such analyses would require a greater degree of uniformity across studies to ensure more reliable conclusions. To add, the concept of AI remains broad and vague, making it challenging to define precise inclusion criteria. While we tried to mitigate this limitation by using various MESH terms in our search strategy, the findings may still be somewhat general in scope, and there remains a possibility that some relevant studies were excluded due to variations in terminology and indexing practices across different databases.

## 4.3. Future implications and recommendations

By using AI, healthcare professionals can benefit from a better understanding of new medical devices and software that ensure faster medical care, reduced workloads, and more accurate treatments which improve patients' overall quality of life. The advantages of AI in healthcare extend beyond medical uses by reducing physical, emotional, and workload stress, especially for exhaustive or repetitive tasks. Based on our results, it is suggested that an analysis would be useful in assessing the total cost of these AI technologies taking into account their importance in ensuring the modernization of healthcare organizations. This suggestion is specifically important in countries with low socio-economic profile such as Lebanon, where infrastructure of AI services is still lacking [127]. On the other hand, it would be interesting to assess additional benefits and drawbacks associated with the use of AI technologies in healthcare. For this reason, it would be helpful to carry out a comparative quantitative analysis between countries engaged in this type of versus those that are not, in order to enhance our knowledge and expand the access of healthcare organizations to AI-based technologies. The improvement of the ethical and legal standards for the use of AI will facilitate its adoption in the society while acknowledging potential risks. The findings will be useful for healthcare professionals as well AI engineers, developers, and researchers to improve the circulation and utilization of medical AI. Therefore, governments can significantly play an important role in supporting empirical research and practical applications through the development and dissemination of an AI-specific implementation framework to improve and adapt policies to secure patient data to promote confidentiality. This framework would address key aspects such as building trust, creating explainable solutions, and tackling ethical concerns regarding issues regarding privacy and data protection associated with AI.

## 5. Conclusion

Our scoping review compiles the available evidence on various AI-based support systems that can be integrated into healthcare practice. It gives an overview of the most used methodologies in the AI in healthcare setting as well as their benefits and challenges in Lebanon. Further, despite the few challenges that exist with this type of technology, the results of the different types of AI are promising in the healthcare setting. However, it is still essential to discover whether these benefits outweigh the risks/challenges related to its use. It remains also essential to consider ethical, legal and privacy concerns as well as ensure that AI is used to enhance the role of healthcare professionals rather than replace them.

## Acknowledgements

Not applicable

## Author contributions

**Conceptualization:** TAWIL Samah, MERHI Samar.

**Data curation:** TAWIL Samah.

**Formal analysis:** TAWIL Samah, MERHI Samar.

**Methodology:** TAWIL Samah.

**Writing – original draft:** TAWIL Samah.

**Writing – review & editing:** TAWIL Samah, MERHI Samar.

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
