## [Decision Letter · Decision Letter 0]

13 Jan 2025

PONE-D-24-54151Investigating the Key Trends in Applying Artificial Intelligence to Health Technologies: A Scoping ReviewPLOS ONE

Dear Dr. Tawil,

Thank you for submitting your manuscript to PLOS ONE. After careful consideration, we feel that it has merit but does not fully meet PLOS ONE’s publication criteria as it currently stands. Therefore, we invite you to submit a revised version of the manuscript that addresses the points raised during the review process.

We look forward to receiving your revised manuscript.

Kind regards,

Justyna Żywiołek

Academic Editor

PLOS ONE

2. We note that your Data Availability Statement is currently as follows: [All relevant data are within the manuscript and its Supporting Information files.] Please confirm at this time whether or not your submission contains all raw data required to replicate the results of your study. Authors must share the “minimal data set” for their submission. PLOS defines the minimal data set to consist of the data required to replicate all study findings reported in the article, as well as related metadata and methods (https://journals.plos.org/plosone/s/data-availability#loc-minimal-data-set-definition).

Additional Editor Comments (if provided):

Reviewers' comments:

Reviewer's Responses to Questions

**Comments to the Author**

1. Is the manuscript technically sound, and do the data support the conclusions?

Reviewer #1: Partly

Reviewer #2: Yes

2. Has the statistical analysis been performed appropriately and rigorously? 

Reviewer #1: No

Reviewer #2: Yes

3. Have the authors made all data underlying the findings in their manuscript fully available?

Reviewer #1: Yes

Reviewer #2: Yes

4. Is the manuscript presented in an intelligible fashion and written in standard English?

Reviewer #1: Yes

Reviewer #2: Yes

5. Review Comments to the Author

Reviewer #1: Dear Authors,

The article is very appropriate considering the importance of the field of artificial intelligence

The article abstract does not mention the objectives, method, and data collection method well

The results were reviewed in the abstract.

In the method:

The method should be clearly specified and its parts should be specified

The prism table should be specified in it

The findings in this study and its classification are appropriate

State the limitations if any

In the method, the number of reviews and independent person is three. What is the role of the third person?

Reviewer #2: The manuscript describes a technically sound piece of scientific research with data that supports the conclusion. Methodology is quite detailed and covers everything. I recommend that the article should be copy-edited by a professional to address minor grammatical, punctuation and typo mistakes (highlighted later in each section).

Use Clear, Concise Language: Avoid overly complex sentences and jargon. Opt for clear, direct phrasing. (Added few examples in the review document)

Summerize the limitations and future recommendations with clear directions.

Results

Condense Descriptive Language: Minimize excessive narrative and refer to tables for detailed figures.

- Example: Instead of “A% of articles in B field, X% in Y field and ....." consider summarizing with “X% of articles have been found in Oncology (see Table 2 for details).”

6. PLOS authors have the option to publish the peer review history of their article (what does this mean? ). If published, this will include your full peer review and any attached files.

**Do you want your identity to be public for this peer review?** For information about this choice, including consent withdrawal, please see our Privacy Policy .

Reviewer #1: No

Reviewer #2: No

---

## [Author Response · Author response to Decision Letter 1]

19 Feb 2025

Dear Editor,

On behalf of the authors, I would like to thank you for the reviewers’ comments. Please find below detailed responses to each of the addressed comments:

Comments Responses

Journal Requirements

Please ensure that your manuscript meets PLOS ONE's style requirements, including those for file naming - Requirements reviewed and amended as per the journal requirement

We note that your Data Availability Statement is currently as follows: [All relevant data are within the manuscript and its Supporting Information files.] Please confirm at this time whether or not your submission contains all raw data required to replicate the results of your study. Authors must share the “minimal data set” for their submission. PLOS defines the minimal data set to consist of the data required to replicate all study findings reported in the article, as well as related metadata and methods (https://journals.plos.org/plosone/s/data-availability#loc-minimal-data-set-definition).

- All relevant data are within the manuscript and its Supporting Information files (a supporting file was included to give an idea about the studies selected)

Reviewers’ comments

Reviewer 1

Overall, I recommend that the article should be copy-edited by a professional to address minor grammatical, punctuation and typo mistakes - Comment addressed and manuscript was reviewed and copy-edited as required

Abstract: - The aims and objectives should be defined more clearly. For instance, rather than stating, “This study presents a scoping review of …,” the authors should elaborate on the necessity of this study - Comment addressed and modification done as requested

Introduction: The introduction is well-structured and provides a clear research question and objectives. However, some refinements are suggested: Line 63: In the sentence, “Certain devices, utilizing an interdisciplinary approach …,” the authors should provide examples or names of these devices and include appropriate citations - Comments addressed and modification done as requested

Line 73: The sentence, “AI has become an indispensable tool in various medical applications, revolutionizing traditional practices,” should be removed as it reiterates content already covered in previous sentences. - Sentence removed as per suggestion

Line 74: The sentence, “Several studies exemplify the integration of artificial intelligence into medicine,” appears overstated and adds little value. Consider removing it. - Sentence removed as per suggestion

Line 75: Sentence "artificially intelligent computer systems inpatient diagnosis" likely contains a typo. It should read "artificially intelligent computer systems in patient diagnosis." - Sentence was removed as per suggestion since it doesn’t fit anymore

Line 87: The sentence, “The Middle East and North Africa region acknowledged the vital importance of adapting…..” is quite lengthy. For improved readability, it can be split into two sentences. For example: “The Middle East and North Africa Region acknowledged the vital importance of adapting to new technologies. They recognized the key role of these technologies in transforming the region and advancing to the forefront of the digital economy and healthcare.” - Sentence corrected as per your suggestion

Line 97: Discussion of Lebanon’s setbacks during COVID-19 can be made more concise by removing jargon - Jargon removed and sentence was reformulated

Line 110: There is a minor grammatical error,

it should be “What AI methodologies have been applied for healthcare systems in Lebanon?" - Sentence corrected as per your suggestion

Materials and Methods:

Line 117-118: Ensure a consistent format for the year range filter, such as "2020 to 2024" or "January 2020 to April 2024." - Sentence corrected as per your suggestion

In the section “Publication Characteristics,” it is mentioned that studies were selected from 2020 to 2023. For clarity, use a clear and consistent range for reviewers and readers. - The mistake was corrected as per your suggestion

Line 118: Replace “following keywords” with “these keywords” for technical accuracy. - Sentence corrected as per your suggestion

Lines 127-130: The sentence discussing the choice of scoping review over systematic review could be rephrased to present a more positive perspective. For instance: "A scoping review was chosen over a systematic review because it allows for a broader exploration of the literature and identification of key concepts and research gaps, which is more suitable for this study given the heterogeneity of the included studies." - Paragraph amended as per your suggestion. Thank you for the rephrase.

Line 149: Replace the typo "v7" with "(7)." - Typo error was corrected as per your suggestion

Line 167: The sentence, “Main findings from the studies were reviewed using descriptive and analytical methods based on different variables outcomes,” should specify the variables for clarity. - Variables were specified and more detailed

Results:

Line 187: The sentence, “The most common reasons for excluding some journal articles after full-text review …,” could be made more specific.

For example: "The most common reasons for excluding articles after full-text review included lack of relevance to the research question and AI not being the primary focus or methodology." - Paragraph amended and corrected as per your suggestion. Thank you for the rephrase.

Line 231: There is a discrepancy between the number of articles per quartile mentioned in the text and Table 2. This should be reconciled - The quartiles were reviewed and the mistake was corrected

Line 244: Replace “didn't” with “did not” to align with academic writing standards. - Correction made as per your suggestion

Lines 253-255: Clarify how many studies fall under each domain to ensure better understanding for the general audience - The number of studies that fall under each category is specified in the sentence following this statement. Percentages were added to make the figure clearer for the audience.

Discussion:

Lines 376-382: Concise this paragraph to 2-3 sentences - Paragraph amended and rephrased according to you recommendation

Lines 388-391: This paragraph could benefit from restructuring for smoother flow.

For example: "Our findings indicate that AI applications in healthcare are predominantly focused on diagnosing and predicting diseases such as cancer, cardiovascular diseases, infectious diseases, and neurological disorders. This focus enhances patient care by allowing healthcare professionals to spend more time with patients, adopt a holistic care approach, and improve patient satisfaction - Paragraph was restricted according to your suggestion. Thank you for the rephrase

Lines 422-423: Strengthen the sentence by briefly summarizing the specific challenges reported in the cited studies. - Sentence amended as per your suggestion

Conclusion:

Line 475: Specify that the study focuses on Lebanon rather than referring to an unspecified Middle Eastern country. - Corrected as per your suggestion

Reviewer 2

The purpose of the study should be specified

The method should be modified, for example, the names of the databases should be included:

Databases: Scopus, PubMed, CINAHL, PsycINFO

Based on what time efficiency the search was performed

Review method

The number of articles should be included in the findings and the characteristics of the number of articles

The findings should be better explained

- The purpose was added. Methods were modified and databases were included. Time efficacy was also added. The number of articles were included and the findings were better explained according to your suggestion

In the study, it refers to the O'Malley method. Based on what method or standard did you proceed and in how many stages?

- Each of the research question specified at the end of the “Introduction” section was answered in the results and discussion sections. As for the methods, searching for relevant studies, selecting studies, and charting the data were detailed in the methods section whereas collating data, summarizing it, and reporting the results are detailed and tabulated in the results section.

Review steps should be written down precisely

- Steps were revised and reformulated more precisely as per your suggestion

It would be better to include it in the table.

- All the 68 studies were added to the table. Specific number of studies obtained from different databases was added to Table 1 as per your suggestion

In the method, the number of reviews and independent person is three. What is the role of the third person?

- To ensure methodological rigidity, a third auditor was consulted in case of any discrepancies, and a consensus on article eligibility was reached through rechecking the information. A more detailed explication was added to the text manuscript

Insert Prism table - Flow diagram is already presented in Figure 1.

Title should be included in tables.

- Titles were added and the table was amended as per your suggestion

Limitations - The limitations section was revised and amended as per your suggestion

---

## [Decision Letter · Decision Letter 1]

18 Mar 2025

Investigating the Key Trends in Applying Artificial Intelligence to Health Technologies: A Scoping Review

PONE-D-24-54151R1

Dear Dr. Tawil,

We’re pleased to inform you that your manuscript has been judged scientifically suitable for publication and will be formally accepted for publication once it meets all outstanding technical requirements.

Kind regards,

Justyna Żywiołek

Academic Editor

PLOS ONE

Additional Editor Comments (optional):

Reviewers' comments:

Reviewer's Responses to Questions

**Comments to the Author**

1. If the authors have adequately addressed your comments raised in a previous round of review and you feel that this manuscript is now acceptable for publication, you may indicate that here to bypass the “Comments to the Author” section, enter your conflict of interest statement in the “Confidential to Editor” section, and submit your "Accept" recommendation.

Reviewer #3: (No Response)

2. Is the manuscript technically sound, and do the data support the conclusions?

Reviewer #3: Yes

3. Has the statistical analysis been performed appropriately and rigorously? 

Reviewer #3: Yes

4. Have the authors made all data underlying the findings in their manuscript fully available?

Reviewer #3: Yes

5. Is the manuscript presented in an intelligible fashion and written in standard English?

Reviewer #3: Yes

6. Review Comments to the Author

Reviewer #3: All points required for the structural composition of a scoping review were met. The composition of the text is acceptable for publication and the theme has profound relevance to the research area.

7. PLOS authors have the option to publish the peer review history of their article (what does this mean? ). If published, this will include your full peer review and any attached files.

**Do you want your identity to be public for this peer review?** For information about this choice, including consent withdrawal, please see our Privacy Policy .

Reviewer #3: No

---

## [Editor Report · Acceptance letter]

PONE-D-24-54151R1

PLOS ONE

Dear Dr. Samah,

I'm pleased to inform you that your manuscript has been deemed suitable for publication in PLOS ONE. Congratulations! Your manuscript is now being handed over to our production team.

Kind regards,

on behalf of

Dr. Justyna Żywiołek

Academic Editor

PLOS ONE